# Solving Homogeneous and Heterogeneous Cooperative Tasks with Greedy Sequential Execution

**Shanqi Liu[1], Dong Xing[1], Pengjie Gu[2], Xinrun Wang[2*], Bo An[2,3*], Yong Liu[1*]**
[1]Zhejiang University  [2]Nanyang Technological University  [3]Skywork AI, Singapore
{shanqiliu,dongxing}@zju.edu.cn, yongliu@iipc.zju.edu.cn,
{pengjie.gu, xinrun.wang, boan}@ntu.edu.sg

## Abstract

Cooperative multi-agent reinforcement learning (MARL) is extensively used for solving complex cooperative tasks, and value decomposition methods are a prevalent approach for this domain. However, these methods have not been successful in addressing both homogeneous and heterogeneous tasks simultaneously which is a crucial aspect for the practical application of cooperative agents. On one hand, value decomposition methods demonstrate superior performance in homogeneous tasks. Nevertheless, they tend to produce agents with similar policies, which is unsuitable for heterogeneous tasks. On the other hand, solutions based on personalized observation or assigned roles are well-suited for heterogeneous tasks. However, they often lead to a trade-off situation where the agent's performance in homogeneous scenarios is negatively affected due to the aggregation of distinct policies. An alternative approach is to adopt sequential execution policies, which offer a flexible form for learning both types of tasks. However, learning sequential execution policies poses challenges in terms of credit assignment, and the limited information about subsequently executed agents can lead to sub-optimal solutions, which is known as the relative over-generalization problem. To tackle these issues, this paper proposes Greedy Sequential Execution (GSE) as a solution to learn the optimal policy that covers both scenarios. In the proposed GSE framework, we introduce an individual utility function into the framework of value decomposition to consider the complex interactions between agents. This function is capable of representing both the homogeneous and heterogeneous optimal policies. Furthermore, we utilize greedy marginal contribution calculated by the utility function as the credit value of the sequential execution policy to address the credit assignment and relative over-generalization problem. We evaluated GSE in both homogeneous and heterogeneous scenarios. The results demonstrate that GSE achieves significant improvement in performance across multiple domains, especially in scenarios involving both homogeneous and heterogeneous tasks.

## 1 Introduction

Centralized training with decentralized execution (CTDE) provides a popular paradigm for value-based cooperative multi-agent reinforcement learning (MARL), which has been extensively employed to learn effective behaviors in many real-world tasks from agents' experiences (Sunehag et al., 2017; Rashid et al., 2018). These tasks encompass different types of scenarios, including homogeneous scenarios where agents are required to take similar actions, e.g., bimanual manipulation (Lee et al., 2013; Caccavale et al., 2008), and heterogeneous scenarios where agents are required to take distinct actions, e.g., autonomous driving through a crossroad (Chen et al., 2021). Following the Individual Global Maximum (IGM) principle (Hostallero et al., 2019), these value decomposition methods can learn centralized value functions as monotonic factorizations of each agent's utility function and enable decentralized execution. Meanwhile, as the parameters of the utility network can be shared among all agents (Gupta et al., 2017), the number of parameters to be trained can be significantly

---

[*]Co-corresponding Authors

reduced. These features together increase the potential of previous CTDE solutions to be applied in large-scale scenarios involving either homogeneous or heterogeneous tasks.

Assuming that scenarios only involve either homogeneous or heterogeneous tasks is oversimplified, as many real-world scenarios require agents to learn both tasks simultaneously (for example, running a restaurant requires waiters and cooks to cooperate within each group and between these two groups (Knott et al., 2021)). However, existing value decomposition methods have not been successful in addressing both types of tasks simultaneously. In homogeneous scenarios, the monotonic value function restricts the value function to sub-optimal value approximations in environments with non-monotonic payoffs (Wang et al., 2020a; Son et al., 2019), they cannot represent the policy that an agent's optimal action depends on actions from other agents. This problem, known as the relative overgeneralization (Panait et al., 2006), prevents the agents from solving all kinds of homogeneous tasks. Recent methods have attempted to address this issue by encouraging agents to simultaneously take the same cooperative actions to find optimal policies in non-monotonic payoffs (Rashid et al., 2020; Mahajan et al., 2019). However, while agents acquiring similar policies can be advantageous for learning in non-monotonic homogeneous scenarios, it impedes the ability of agents to adapt to heterogeneous scenarios. Meanwhile, in heterogeneous scenarios, one of the main challenges is obtaining distinct policies among agents when the utility network is shared among all of them. This shared utility network makes it difficult for agents to learn and exhibit diverse policies that are necessary for such scenarios. Therefore, several methods employ techniques such as incorporating agent ID as input to construct different policies (Li et al., 2021a) or assigning different roles to encourage diverse behaviors (Christianos et al., 2021; Wang et al., 2020c). Nevertheless, these methods still encounter other challenges. They tend to result in fixed policies that can only represent a single solution mode of the task, which precludes cooperation when working with dynamically changing agents (Fu et al., 2022). In addition, simply aggregating distinct policies results in a trade-off scenario, where performance in homogeneous scenarios is negatively impacted (Christianos et al., 2021; Li et al., 2021a).

To address these challenges, sequential execution policies have been introduced. These policies allow agents to take actions based on the actions of previous agents, enabling the learning of both homogeneous and heterogeneous tasks (Fu et al., 2022; Liu et al., 2021). In this approach, as latter agents can adjust their actions based on the actions of earlier agents, they can exhibit either homogeneous or heterogeneous policy to cooperate with the previous agents, depending on the specific situation. However, sequential execution methods encounter challenges in credit assignment, as the policy form does not conform to the IGM principle, precluding current value decomposition methods from learning each agent's individual utility. Additionally, as the former executed agents lack action information about the latter agents, the former executed policies may still converge to a sub-optimal solution and cannot solve the relative overgeneralization problem thoroughly.

In this work, we propose Greedy Sequential Execution (GSE) which is capable of addressing these problems and adapting to both homogeneous and heterogeneous tasks. Specifically, we first propose a value decomposition scheme that captures the interactions between agents while adhering to the IGM principle. This value decomposition enables agents to learn individual utilities that take into account interactions with all other cooperative agents. We demonstrate that such individual utilities can accurately represent both homogeneous and heterogeneous payoff matrices. However, since the individual utilities require the actions of all other agents to conduct actions, which is infeasible in sequential execution, they cannot be directly used as the policy's value function. To address this issue, we further propose an explicit credit assignment method that calculates a greedy marginal contribution as the credit value for each agent's policy in sequential execution. The insight behind this is that each agent's optimal cooperative policy is to maximize the marginal contribution of joining the group of previously executed agents while considering that the latter agents will take optimal cooperative actions to cooperate with it. We show that the greedy marginal contribution can overcome the relative over-generalization problem by avoiding taking conservative actions that lead to mis-coordination. Furthermore, using the explicit credit assignment method can address the challenges of learning each agent's individual utility in the sequential execution policy. It allows for the precise allocation of credit to each agent's actions, enabling effective learning of the sequential execution policy. We evaluated GSE in comparison to several state-of-the-art baselines in various scenarios, including homogeneous tasks, heterogeneous tasks, and a combination of both tasks. The results demonstrate that our proposed method achieves a significant improvement in performance

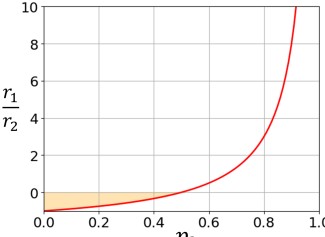 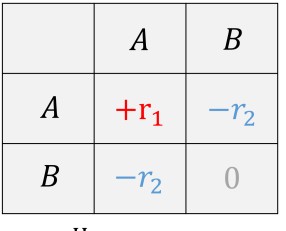 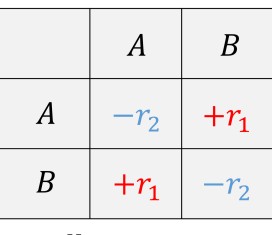

Homogeneous              Heterogeneous

Figure 1: Left: The illustration of Eq. (4) and Eq. (5). The orange area is where the methods using $Q_i(\tau_i, a_i)$ as the individual utilities have the maximum possibility to solve both tasks depicted by the example payoff matrixes. Middle: Non-monotonic payoff matrix of the homogeneous example, where agents are required to take action $A$ simultaneously to achieve cooperation and taking action $A$ alone results in a penalty. Right: Payoff matrix of the heterogeneous example, where taking the same action is punished and taking distinct action is rewarded.

across all domains and the most prominent advantages in scenarios involving both tasks, while other methods struggle to provide effective solutions.

## 2 PRELIMINARIES

**Dec-POMDP.** A fully cooperative multi-agent sequential decision-making task can be described as a decentralized partially observable Markov decision process (Dec-POMDP), which is defined by a set of possible global states $S$, actions $A_1, ..., A_n$, and observations $\Omega_1, ..., \Omega_n$. At each time step, each agent $i \in \{1, ..., n\}$ chooses an action $a_i \in A_i$, and they together form a joint action $\mathbf{u} \in U$. The next state is determined by a transition function $P : S \times U \rightarrow S$. The next observation of each agent $o_i \in \Omega_i$ is updated by an observation function $O : S \rightarrow \Omega$. All agents share the same reward $r : S \times U \rightarrow \mathbf{r}$. The objective is to learn a local policy $\pi_i(a_i|s_t)$ for each agent such that they can cooperate to maximize the expected cumulative discounted return $R_t = \sum_{j=0}^{\infty} \gamma^j r_{t+j}$. The joint value function is $Q_{tot} = E_{s_{t+1}:\infty, a_{t+1}:\infty}[R_t | s_t, \mathbf{u_t}]$. The observation of each agent can also be replaced by the history of actions and observations of each agent to handle partial observability (Sunehag et al., 2017; Rashid et al., 2018). The history of actions and observations of agent $i$ can be viewed as $\tau_i$ which is $(o_i^0, a_i^0, ..., o_i^t)$.

**Value Decomposition Methods.** Current value decomposition methods represent the joint action value function $Q_{tot}$ as a mixing of per-agent utility functions to form the CTDE structure, where Individual Global Max (IGM) principle (Hostallero et al., 2019) is wildly used to enable efficient decentralized execution:

$$\arg\max_u (Q_{tot}(s, u)) = \{\arg\max_{a_1}(Q_1(\tau_1, a_1)), ..., \arg\max_{a_n}(Q_n(\tau_n, a_n))\}. \tag{1}$$

QMIX (Rashid et al., 2018) combines the agent utilities via a continuous monotonic function to satisfy IGM, i.e.,

$$Q_{tot}^{QMIX}(s, u) = f(Q_1(\tau_1, a_1), ..., Q_n(\tau_n, a_n))$$
$$\frac{\partial Q_{tot}^{QMIX}}{\partial Q_i} > 0, \forall i \in n. \tag{2}$$

**Shapley Value and Marginal Contribution.** We introduce the concept of the marginal contribution of Shapley Value (Shapley, 2016). The marginal contribution of Shapley Value for agent $i$ is defined as

$$\phi_i = v(C) - v(C/i) \tag{3}$$

where $C$ is a team of agents that cooperate with one another to achieve a common goal, and $C/i$ represents the set in the absence of agent $i$. $v(C)$ refers to the value function that estimates the cooperation of a set of agents.

## 3 MOTIVATING EXAMPLES

In this section, we utilize two example tasks as motivating examples to illustrate the advantages and limitations of various methods. Their payoff matrices are depicted in Figure 1.

### 3.1 ISSUES OF CURRENT METHODS

We investigate the limitations of current value decomposition methods by analyzing the form of individual utility. Currently, these methods model the individual utility of each agent as a value function $Q_i(\tau_i, a_i)$ to learn the decentralized policy. However, since the returns of both the homogeneous and heterogeneous tasks depend on the actions of other agents, such an individual utility is not sufficient to represent the cooperation policy. We propose a more comprehensive individual utility function, $Q_c^i(\tau_i, u_i^-, a_i)$, where $u_i^-$ represents the joint actions of all other agents who have the potential to cooperate (discussed in detail in Section 4.1). According to this decomposition, the individual utility $Q_i(\tau_i, a_i)$ can be viewed as a variable sampled from the distribution $Q_c^i(\tau_i, u_i^-, a_i)$ over $u_i^-$. This understanding enables us to demonstrate that $Q_i(\tau_i, a_i)$ cannot represent homogeneous and heterogeneous policies simultaneously, resulting in the trade-off when learning both types of tasks concurrently. We illustrate this through the two motivating example scenarios.

For the two example tasks, the learned policy fails to represent the optimal policy when

$$\frac{r_1}{r_2} < \frac{2p_b - 1}{1 - p_b}. \tag{4}$$

In the homogeneous scenarios, while the learned policy can never represent the optimal policy in the heterogeneous scenarios and the possibility $P_c$ of achieving cooperation is

$$P_c = 2 \cdot p_b \cdot (1 - p_b). \tag{5}$$

where $p_b$ is the probability of each policy taking action $B$. The detailed derivation and proof are included in Appendix 5. Figure 1 illustrates the result of Eq. (4) and Eq. (5). The result indicates that as the $p_b$ grows the $\frac{r_1}{r_2}$ grows exponentially in the homogeneous scenarios. Therefore, we notice that there is a trade-off as solving the homogeneous non-monotonic matrixes requires the $p_b$ to decrease to zero, while solving the heterogeneous matrixes needs to increase $p_b$ when $p_b$ is below 0.5. As a result, the ability of these methods to effectively learn both homogeneous and heterogeneous tasks simultaneously is limited.

### 3.2 SEQUENTIAL EXECUTION AS A REMEDY

Another method is proposed that models individual utility as $Q_s^i(\tau_i, a_{1:i-1}, a_i)$, which is a sequential execution policy. We illustrate that as $p_b$ serves as a known variable for the subsequently executed agent, the utility of the latter agent can choose actions in accordance with the actions of the former agent, thereby achieving the desired similar or distinct policies. As a result, the overall policy can encompass both homogeneous and heterogeneous policies. Although the interactions between agents involve communication, the bandwidth is limited as the actions are one-hot vectors. Therefore, these methods retain the potential to be implemented in complex real-world scenarios. However, the individual utility $Q_s^i(\tau_i, a_{1:i-1}, a_i)$ does not satisfy the IGM principle as the former agents' utilities lack the necessary information about other agents' actions (detailed in the Appendix 6), which precludes implicit credit assignment. Additionally, the individual utility of the former agent remains $Q_s^i(\tau_i, a_i)$, which encounters the problem of relative over-generalization. Therefore, while the policy form is capable of representing the target policy mode, it is not guaranteed to converge to it.

## 4 GREEDY SEQUENTIAL EXECUTION

In this section, we propose Greedy Sequential Execution (GSE) to address the problems of credit assignment and relative over-generalization in the sequential execution policy. Specifically, we first propose a value decomposition that can capture the interactions between agents while adhering to the IGM principle. Then, we propose an actor-critic structure based on the value decomposition that trains a critic and calculates a greedy marginal contribution as credit value for sequential execution policy training. This explicit credit value addresses the credit assignment problem in learning

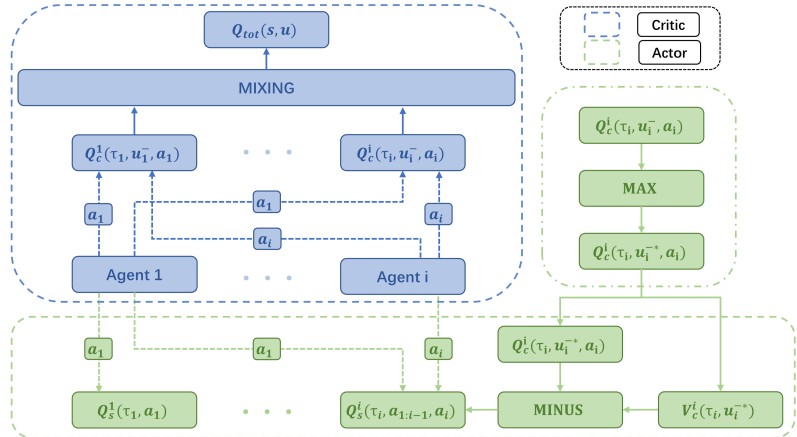

Figure 2: The architecture of our method. Upper (Blue): The architecture for critic $Q_c^i(\tau_i, u_i^-, a_i)$ that we elaborated in Section 4.1. Lower (Green): The framework of the calculation of the greedy marginal contribution based on $Q_c^i(\tau_i, u_i^{-*}, a_i)$ and the sequential execution policy $Q_s^i(\tau_i, a_{1:i-1}, a_i)$.

sequential execution policy. Meanwhile, the greedy value of marginal contribution tackles the relative overgeneralized problem of former executed agents in sequential execution. As a result, GSE achieves optimal cooperation in scenarios involving both homogeneous and heterogeneous tasks by effectively learning the sequential execution policy.

## 4.1 VALUE DECOMPOSITION

Since the current individual utility $Q_i(\tau_i, a_i)$ is insufficient to represent the optimal policy in both homogeneous and heterogeneous scenarios, we illustrate that $Q_c^i(\tau_i, u_i^-, a_i)$ is a more comprehensive utility which can capture the interactions between agents.

**Theorem 4.1.** *For any $r_{tot}(s, u)$, the corresponding $Q_{tot}(s, u) = \mathbb{E}\left[\sum_{t=0}^{\infty} \gamma^t r_{tot}(s, u) \mid \pi\right]$ and each agent's utility $Q_c^i(\tau_i, u_i^-, a_i)$ satisfies*

$$\arg\max_u(Q_{tot}(s, u)) = \{\arg\max_{a_1}(Q_c^1(\tau_1, u_1^-, a_1)), ..., \arg\max_{a_n}(Q_c^n(\tau_n, u_n^-, a_n))\}. \quad (6)$$

Detailed proof can be found in Appendix 4. Theorem 4.1 indicates that the value decomposition using utility $Q_c^i(\tau_i, u_i^-, a_i)$ can represent the value decomposition given any reward function and satisfies the IGM principle. Therefore, we can use all $Q_c^i$ to calculate $Q_{tot}$ through a monotonic mixing network similar to QMIX, and Theorem 4.1 illustrates the mixing value is unbiased. Specifically, the overall critic value function consists of each agent's adaptive utility $Q_c^i(\tau_i, u_i^-, a_i)$ and a mixing network to produce the global Q-value $Q_{tot}(s, u)$. The critic value function is learned by optimizing

$$\begin{aligned} \mathcal{L}_{TD}(\theta) &= \mathbb{E}_\pi[Q_{tot}(s_t, \mathbf{u}_t) - y_t]^2 \\ y_t &= r_t + \gamma \max_{\mathbf{u}_{t+1}} Q_{tot}(s_{t+1}, \mathbf{u}_{t+1}). \end{aligned} \quad (7)$$

## 4.2 CREDIT ASSIGNMENT VIA GREEDY MARGINAL CONTRIBUTION

The utilization of the critic value function as the agent policy's value function is not feasible due to its reliance on the actions of all other agents, resulting in a deadlock situation. An alternative approach is the utilization of a sequential execution policy, represented by $Q_s^i(\tau_i, a_{1:i-1}, a_i)$, which allows for the consideration of the actions of former agents. However, this approach does not adhere to the principles of IGM principal and encounters the relative over-generalization problem. To overcome these limitations, we propose an explicit credit assignment method utilizing marginal contribution of Shapley values for the learning of a policy capable of addressing both homogeneous and heterogeneous tasks.

According to the sequential execution process, each agent takes actions with the actions of former executed agents, which is equivalent to agent $i$ joining a group consisting of former executed agents. Therefore, the policy represented by $Q_s^i(\tau_i, a_{1:i-1}, a_i)$ should aim to maximize the marginal contribution of agent $i$ joining the group of former executed agents. Based on the critic value function, the marginal contribution of agent $i$ can be calculated as

$$\phi_i(\tau_i, a_{1:i-1}, a_i) = v(T_i) - v(T_i/i) = Q_c^i(\tau_i, a_{1:i-1}, a_i) - V_c^i(\tau_i, a_{1:i-1}), \qquad (8)$$

where $\phi_i$ is the marginal contribution. The reason behind this is that the uniqueness of the optimal action within the entire action space, where the primary actions often result in mis-coordination. Therefore, we can use $V_c^i(\tau_i, a_{1:i-1})$ as the value of agent $i$ not joining the former group to calculate the marginal contribution. However, such a calculation still faces other problems. Since the critic value function $Q_c^i$ is trained by taking $u_i^-$ instead of $a_{1:i-1}$ as input, the accuracy of the calculated marginal contribution may be affected. Additionally, such a marginal contribution encounters the relative over-generalization problem, as the value of action $a_i$ depends on overall joint actions $u_i^-$ and the current marginal contribution cannot consider the actions of latter agents $a_{i+1:n}$, leading the marginal contribution of $a_i$ converge to an average value as we discussed in Section 3. To address these problems, we propose a greedy marginal contribution,

$$\phi_i^*(\tau_i, a_{1:i-1}, a_i) = Q_c^i(\tau_i, a_{1:i-1}, a_{i+1:n}^*, a_i) - V_c^i(\tau_i, a_{1:i-1}, a_{i+1:n}^*), \qquad (9)$$

where $a_{i+1:n}^*$ is the optimal cooperative actions to cooperate with former agents that maximize $\phi_i$. This approach ensures that the marginal contribution accurately represents the potential optimal value of action $a_i$, rather than an average value, thus addressing the issue of relative over-generalization. Furthermore, by including full information about $u_i^-$ as inputs, this approach allows for the accurate calculation of values by $\phi_i$. However, $a_{i+1:n}^*$ is not directly observable. Intuitively, the greedy marginal contribution means that each agent takes the action under the condition that all the latter agents will take the optimal cooperative action to cooperate with it. Therefore, we use the greedy actions $a_{i+1:n}^g$ from the behavior policy to represent the $a_{i+1:n}^*$,

$$a_{i+1:n}^g = \{\arg\max_{a_{i+1}}(Q_s^{i+1}(\tau_1, a_{1:i})), ..., \arg\max_{a_n}(Q_s^n(\tau_i, a_{1:n-1}))\}, \qquad (10)$$

However, such an estimation is not guaranteed to be correct when the behavior policy has not converged in the early training period. Therefore, we additionally use the Monte Carlo method to estimate the $a_{i+1:n}^*$ to address this problem. Specifically, we sample $M$ random joint actions as $a_{i+1:n}^{j=1:M}$ and search for the $a_{i+1:n}$ with the maximum value of $\phi_i$ in the collection of $a_{i+1:n}^{j=1:M}$ and $a_{i+1:n}^g$ to be the $a_{i+1:n}^*$.

In this way, we have our sequential execution policy's value function $Q_s^i(\tau_i, a_{1:i-1}, a_i)$ and decentralized policy as $\pi(a_i | \tau_i, a_{1:i-1}) = \arg\max_{a_i}(Q_s^i(\tau_i, a_{1:i-1}, a_i))$. The overall loss is

$$\mathcal{L}_p(\mu) = \mathbb{E}_\pi[Q_s^i(\tau_i, a_{1:i-1}, a_i) - \phi_i^*(\tau_i, a_{1:i-1}, a_i)]^2. \qquad (11)$$

In terms of practical implementation, we utilize an attention structure to address the varying dimensions of $a_{1:i-1}$, enabling the implementation of parameter sharing. The overall structure of our method is shown in Figure 2.

## 5 UNDERSTANDING TASK TYPES THROUGH MULTI-XOR GAMES

To evaluate how the different task types affect the performance of all methods, we devise a simple one-step randomized Multi-XOR game. Agents in the game can either take a cooperative action $C$ or a lazy action $L$. The game requires two out of four agents to take cooperative actions $C$ simultaneously to achieve successful cooperation. In contrast, if a single agent takes cooperative actions $C$, a homogeneous penalty is applied, and when more than two agents take cooperative actions $C$, a heterogeneous penalty is given. To prevent agents from learning a fixed strategy, we randomly selected a dummy agent at the beginning of each episode, which cannot participate in cooperation and can only take the lazy action $L$. We set three scenarios according to the challenges of tasks, the homogeneous scenario with only the homogeneous penalty, the heterogeneous with only the heterogeneous penalty, and the mixing scenario with both penalties. These methods that we compared include CDS (Li et al., 2021a), which employs mutual information to learn agent ID-specific policies; Shapley (Li et al., 2021b), which utilizes Shapley Value to estimate complex agent interactions; and

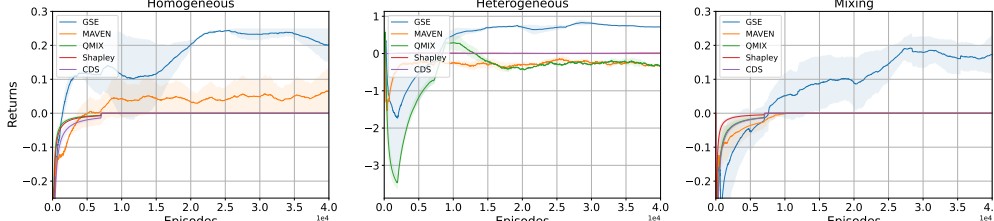

Figure 3: Results of Multi-XOR game. Left: Homogeneous challenge. Middle: Heterogeneous challenge. Right: Mixing scenario that combines both the Homogeneous and Heterogeneous challenges.

both MAVEN (Mahajan et al., 2019) and QMIX (Rashid et al., 2018), which are common value decomposition methods. For a detailed introduction to all the methods compared, please refer to Appendix 1. Detailed reward settings are included in Appendix 2.

The results displayed in Figure 3 indicate that our method effectively masters both homogeneous and heterogeneous tasks in the randomized Multi-XOR game. Specifically, in the homogeneous scenario, all other methods fail to overcome the non-monotonicity and learn a lazy policy that never engages in cooperative action to avoid penalties, except for MAVEN which is proposed to learn cooperative policy in non-monotonic environments. In the heterogeneous scenario, MAVEN and QMIX fail to learn heterogeneous policies and take cooperative action together, resulting in failure. In the mixing scenario, our method also outperforms other methods, indicating its robustness in handling both homogeneous and heterogeneous tasks simultaneously. Overall, our method demonstrates superior performance in adapting to different types of cooperative tasks compared to other methods.

# 6 EXPERIMENTS

## 6.1 EXPERIMENTAL SETTINGS

We have developed a series of challenging cooperative scenarios that involve the integration of both homogeneous and heterogeneous tasks to evaluate all methods. The experiments are conducted based on MAgent (Zheng et al., 2018) and Overcooked (Sarkar et al., 2022). We implement five tasks in MAgent: *lift*, *heterogeneous_lift*, *multi_target_lift*, *pursuit* and *bridge*. These scenarios can be classified into three categories: homogeneous, heterogeneous, and mixing.

**Homogeneous scenarios:** In our implementation, we have included two homogeneous scenarios within the task of *lift*. Specifically, the *lift* task requires the coordination of two agents to lift a cargo. Successful completion of the task necessitates the simultaneous execution of the "lift" action by both agents, otherwise, the cooperation will fail and the agent who takes the action will incur a penalty of $-r_2$, similar to the homogeneous task discussed in Section 3. To evaluate all methods, we have chosen two scenarios with $-r_2 = 0$ and $-r_2 = -0.3$, as these scenarios represent situations without and with the relative over-generalization problem, respectively.

**Heterogeneous scenarios:** In *heterogeneous_lift*, agents must lift cargos cooperatively with changing partners. Rewards are given for two-agent lifts, with penalties for more than two. Each episode randomly excludes one or two agents from cooperating, preventing fixed policy learning. The challenge of this task lies in the fact that agents must adapt to varied partners for successful lifting. This necessitates the learning of heterogeneous policy to succeed.

**Homogeneous & heterogeneous scenarios:** We also implement three tasks that necessitate both homogeneous and heterogeneous policies to be solved. The first task, *multi_target_lift*, is similar to *lift*, but with different cargos with varying rewards. The challenge of this task is that agents must learn to lift cargos with lower rewards in order to achieve optimal cooperation, instead of all competing to lift cargos with higher rewards. The second task, *pursuit*, requires agents to catch a prey and rewards are only given when two agents attack together, otherwise, a penalty is imposed. Additionally, the prey will be killed if more than two agents attack together, resulting in a significant penalty, thus requiring agents to perform differently to avoid killing the prey. In the third *bridge* task, agents start on opposite sides of the map and must navigate through a single tunnel, initially blocked by an

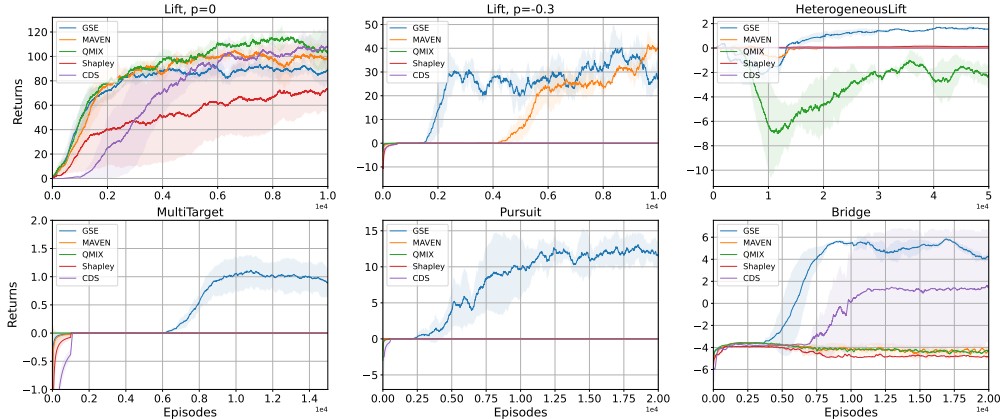

Figure 4: Top: The performance results in *lift* with the penalty dropping from 0 to -0.3 and *heterogeneous_lift*. Bottom: The results in *multi_target_lift*, *pursuit* and *bridge*.

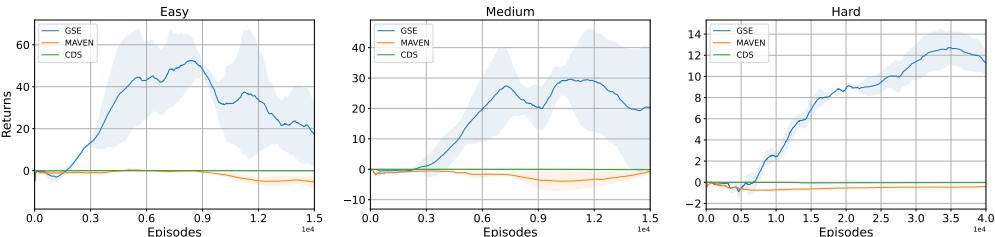

Figure 5: Results of Overcooked. Difficulty increases from left to right.

obstacle. Two agents must cooperate to break this barrier, then learn to retreat, preventing tunnel blockage for others.

We also designed three scenarios in Overcooked that require both homogeneous and heterogeneous policies. In the game, the objective is to cook and deliver soups. We designed the picking of onions to be a cooperative action that requires two agents to work together. However, the cooking and delivering process introduce the need for agents to exhibit different behaviors, such as yielding to others. We evaluated our approach on three maps of different difficulties: an easy small map with multiple onions, a medium-difficulty small map with a single onion, and a challenging large map with a single onion. Please refer to Appendix 2 and Appendix 3 for more experimental settings.

In the experiments, the methods compared are introduced in Section 5. All the baseline methods use the agent ID as an extra input to enable the learning of heterogeneous policies, while our method does not. All methods use the same basic hyperparameters and network structures with similar parameters to ensure the comparison is fair.

## 6.2 PERFORMANCE

We evaluate the performance of various methods in three types of scenarios: homogeneous, heterogeneous, and homogeneous & heterogeneous. The results of MAgent scenarios are shown in Figure 4 and Overcooked are in Figure 5. The results for the homogeneous scenarios indicate that the performance of all comparison methods is heavily influenced by the increase in penalty. Most methods are able to learn an optimal policy when the penalty is zero, but they fail to converge to a cooperative policy when the penalty is -0.3. Their policies converge to a sub-optimal policy that never takes cooperative actions in order to avoid the mis-coordination penalty, except for MAVEN which is proposed to solve the relative overgeneralization problem. In contrast, our method can find the optimal cooperative policy in both scenarios regardless of the growing penalty. This result indicates that our method is able to overcome the relative over-generalization problem.

Figure 6: The first two figures are ablations of specific parts of our method. The third figure is the ablation of sample number $M$, the fourth figure is training with a larger scale of agents.

In the heterogeneous scenario, we observe that QMIX converges to a sub-optimal policy that all agents lift together, resulting in penalties. Similarly, other methods also struggle to learn the cooperative policy but can learn to take lazy actions to avoid penalties. All of these policies reflect a failure to learn distinct policies that can adapt to dynamically changing partners. In contrast, our method demonstrates the ability to adapt to the actions of other agents and outperforms other methods in terms of both final performance and sample efficiency when learning heterogeneous policy.

Lastly, we evaluate all methods in mixing scenarios involving both homogeneous and heterogeneous tasks. The results show that our method has the most significant advantage in these scenarios. In MAgent scenarios, most methods coverage to the conservative policy since the penalty comes from mis-coordination of both homogeneous and heterogeneous actions which they cannot handle simultaneously. However, CDS can solve the *bridge* scenario, which is because the homogeneous behavior only involves breaking the obstacle and most required behaviors are heterogeneous actions. In Overcooked scenarios, we compared our method with MAVEN and CDS as they represent methods that can handle complex homogeneous and heterogeneous tasks, respectively. The results are consistent with other mixing scenarios. Since the learning difficulty of these tasks mainly arises from learning two conflicting tasks modes simultaneously, this result indicates that our method can unify the learning of similar and distinct policies. On the contrary, all other methods struggle to learn an efficient policy to solve the tasks due to their narrow policy representation ability.

### 6.3 ABLATIONS

We conduct several ablation studies to evaluate the effectiveness of our proposed method in homogeneous & heterogeneous scenarios, *multi_target_lift* and *multi_XOR*. Specifically, we evaluate our method training without using greedy actions, meaning we relied on marginal contributions instead of greedy marginal contributions. Additionally, we evaluated the method without using marginal contributions, instead directly using the policy value function to fit the $Q_{tot}$. The results of these evaluations are presented in Figure 6. The results indicate that training without using greedy actions can significantly degrade performance, as using greedy actions helps to overcome the relative over-generalization problem. Training without using marginal contributions also degrades performance in both scenarios as the sequential execution policy does not satisfy the IGM principle, underscoring the importance of our proposed utility function. Additionally, we evaluated how the sample number $M$ affects performance. The results demonstrate that a small value of $M$ can be problematic as it may not select the greedy value of actions. However, a reasonably large value of $M$ is sufficient as increasing $M$ beyond 5 does not further improve performance. Finally, we evaluated our method with a larger number of agents, specifically by using double or triple the number of agents compared to our standard configurations. The results in Figure 6 demonstrate that our method is capable of handling a larger number of agents without affecting performance.

### 7 CONCLUSION

This work introduces Greedy Sequential Execution (GSE), a unified solution designed to extend value decomposition methods for tackling complex cooperative tasks in real-world scenarios. Our method employs a utility function that accounts for complex agent interactions and supports both homogeneous and heterogeneous optimal policies. Furthermore, by implementing a greedy marginal contribution, GSE overcomes the relative over-generalization problem. Our experiments show that GSE significantly improves performance across various domains, especially in mixed-task scenarios.

ACKNOWLEDGEMENTS

This work was supported by NSFC 62088101 Autonomous Intelligent Unmanned Systems and China Scholarship Council.

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

# A  RELATED WORK

Recent works have extended MARL from small discrete state spaces (Yang & Gu, 2004; Busoniu et al., 2008) to high-dimensional, continuous state spaces (Lowe et al., 2017; Peng et al., 2017). The progresses of deep reinforcement learning give rise to an increasing effort in designing general-purpose deep MARL methods for complex multi-agent environments, including COMA (Foerster et al., 2018), MADDPG (Lowe et al., 2017), MAPPO (Yu et al., 2021) and etc. Currently, CTDE is considered to be the de facto mainstream paradigm in this field (Lowe et al., 2017; Iqbal & Sha, 2019). In terms of specific methods, the Value-Decomposition Network (VDN) (Sunehag et al., 2017) utilizes the factorization of joint-action Q-values as the sum of each agent's utility. QMIX (Rashid et al., 2018) is an extension of VDN which allows the joint action Q-value to be a monotonically increasing combination of each agent's utility, which can vary depending on the global state. There are also other variants proposed to extend the applicability of the value decomposition methods. For instance, QPLEX (Wang et al., 2020a) and QTRAN (Son et al., 2019) aim to learn value functions with complete expressiveness capacity. MAVEN (Mahajan et al., 2019) hybridises value and policy-based methods by introducing a latent space for hierarchical control. This allows MAVEN to achieve committed, temporally extended exploration. Weighted QMIX (Rashid et al., 2020) is based on QMIX and rectifies the suboptimality by introducing weights to place more importance on the better joint actions. UneVEn (Gupta et al., 2021) learns a set of related tasks simultaneously with a linear decomposition of universal successor features. Despite the effectiveness of these methods, they are commonly designed to facilitate the learning of similar policies, it can be detrimental to the acquisition of heterogeneous policies.

To solve the heterogeneous tasks, previous methods choose to add agent-specific information to the observation or assign different roles to learn the distinct policies. PSHA (Terry et al., 2020) proposes an agent indication to enable agents to represent heterogeneous policies. CDS (Li et al., 2021a) uses mutual information to learn an agent ID-specific policy to deal with the problem of learning diversity policies. ROMA (Wang et al., 2020c) proposes a role-oriented MARL framework to make agents specialized in certain tasks. However, these methods, which solely focus on learning distinct policies, often come at the cost of sacrificing the advantages associated with learning in homogeneous scenarios. Furthermore, these methods tend to learn fixed policies that lack the necessary flexibility.

Other methods use a sequential execution policy to represent distinct policies. AR (Fu et al., 2022) proposes a centralized sequential execution policy to solve permutation games. MAiF (Liu et al., 2021) uses a sequential execution policy to learn a path-finding and formation policy for a multi-agent navigation task. These methods can represent the optimal policy in both homogeneous and heterogeneous scenarios. However, a naive sequential execution policy is not guaranteed to converge to optimal policy and has the problem of credit assignment. Additionally, there are also methods such as HAPPO (Kuba et al., 2021) that use sequential policy updates to guarantee monotonic policy improvement of PPO (Schulman et al., 2017). MAT (Wen et al., 2022) adopts sequential policy updates within the structure of a transformer. This design is aimed at executing updates both monotonically and in parallel, thereby enhancing the time efficiency compared to previous methods like HAPPO. SeCA (Zang et al., 2023) constructs a new advantage value to improve upon PG-based methods. Different from focusing on an increment of PG-based methods, our work is proposed to extend the applicability of value decomposition methods to solve the mixing of homogeneous and heterogeneous tasks.

Our work is also related to the credit assignment. Previous methods usually use implicit credit assignment methods to learn the policy, such as VDN and QMIX. However, explicit credit assignment methods have also been proposed. For instance, COMA (Foerster et al., 2018) utilizes a counterfactual advantage to learn the value function. Other methods use Shapley Value (Shapley, 2016) as the credit value of each agent. Shapley Value originates from cooperative game theory and is able to distribute benefits reasonably by estimating the contribution of participating agents. In these methods, SQDDPG (Wang et al., 2020b) and Shapley (Li et al., 2021b) use Shapley Value to estimate the complex interactions between agents. However, these methods can only get approximated Shapley value as calculating the Shapley value involves exponential time complexity (Wang et al., 2020b) and they are not designed to learn similar and distinct policies simultaneously. In this work, we introduce an explicit credit assignment method using marginal contribution in Shapley value to learn a sequential execution policy that can represent the optimal policy in scenarios with a mixing of homogeneous and heterogeneous tasks.

# B  SCENARIOS SETTINGS AND TRAINING DETAILS

In the Multi-XOR games, agents receive two types of rewards, as illustrated in Table 1 and 2. Table 1 displays the homogeneous reward, which exhibits a non-monotonic payoff. This poses a challenge of relative overgeneralization for the learning process. Table 2 presents the heterogeneous reward, where agents are required to take distinct actions. Specifically, if two agents choose the joint actions $C\&C$ to solve the task, the other two agents must choose $L\&L$; otherwise, a penalty will be imposed. However, if all agents select $L\&L$, the return will be zero.

In MAgent, each agent corresponds to one grid and has a local observation that contains a square view centered at the agent and a feature vector including coordinates, health point (HP) and ID of agents nearby, and the agent's last action. The discrete actions are moving, staying, and attacking. The global state of MAgent is a mini-map ($6 \times 6$) of the global information. The opponent's policies used in experiments are randomly escaping policy in *pursuit*. We choose five different scenarios *lift*, *heterogeneous_lift*, *multi_target_lift*, *pursuit* and *bridge*. There are detailed settings of these scenarios, as shown in Table 3. We demonstrate the payoff matrix by showing the $R$ as the reward returned when cooperation is achieved, $P_{ho}$ as are penalty when taking cooperative action but failing to achieve cooperation in homogeneous scenarios, and $P_{he}$ as the penalty for taking the same action in heterogeneous scenarios.

|   | C | L |
|---|---|---|
| C | +0.5 | -0.3 |
| L | -0.3 | 0 |

Table 1: Homogeneous payoff matrix of the Multi-XOR game.

|   | C&C | L&L |
|---|---|---|
| C&C | -10 | +0.5 |
| L&L | +0.5 | 0 |

Table 2: Heterogeneous payoff matrix of the Multi-XOR game.

|   | Lift | HeterogeneousLift | MultiTargetLift | Pursuit | Bridge |
|---|---|---|---|---|---|
| Agent number | 3 | 4 | 4 | 4 | 4 |
| Object number | 3 | 1 | 2 | 1 | 0 |
| Map size | $6 \times 6$ | $15 \times 15$ | $12 \times 12$ | $10 \times 10$ | $11 \times 11$ |
| Payoff | R=1,$P_{ho}$=0,-0.3,$P_{he}$=0 | R=1,$P_{ho}$=0,$P_{he}$=-100 | R=0.25,0.5,$P_{ho}$=-0.2,$P_{he}$=-20,-40 | R=0.5,$P_{ho}$=-0.1,$P_{he}$=-1 | R=0.5,$P_{ho}$=-0.03,$P_{he}$=0 |

Table 3: Settings of MAgent Scenarios. $R$ is the reward, $P_{ho}$ is the penalty of mis-coordination of homogeneous behavior, $P_{he}$ is the penalty of mis-coordination of heterogeneous behavior.

In the Overcooked environment, the objective is to perform a series of tasks involving onions, dishes, and soups. The agents are required to place 2 onions in a pot, let them cook for 5 timesteps, transfer the resulting soup into a dish, and finally serve it, which rewards all players with a score of 20. There are six possible actions available to the agents: up, down, left, right, noop (no operation), and interact. Notably, the action of picking up onions requires two agents to simultaneously take the "interact" action, otherwise a penalty of -0.1 is incurred. On the other hand, actions such as putting onions into the pots can be performed by a single agent. To evaluate the difficulty level of different scenarios, we have designed three maps with varying levels of complexity. The easy map consists of more onions and a smaller map size (5×5), making it relatively easier to solve. The medium map, on the other hand, contains a single onion and a smaller map size (5×5). Finally, the hard map poses a greater challenge with its larger map size (7×7) and a single onion, making exploration more demanding for the agents.

We set the discount factor as 0.99 and use the RMSprop optimizer with a learning rate of 5e-4 for policy and 1e-3 for the critic. The $\epsilon$-greedy is used for exploration with $\epsilon$ annealed linearly from 1.0 to 0.05 in 700k steps. The batch size is 4 and updating the target network every 200 episodes. The length of each episode in MAgent is limited to 100 steps in bridge and 50 for others, except for Multi-XOR which is a single-step game. The sample number $M$ of our method is 5 in all scenarios. We run all the experiments five times with different random seeds and plot the mean/std in all the

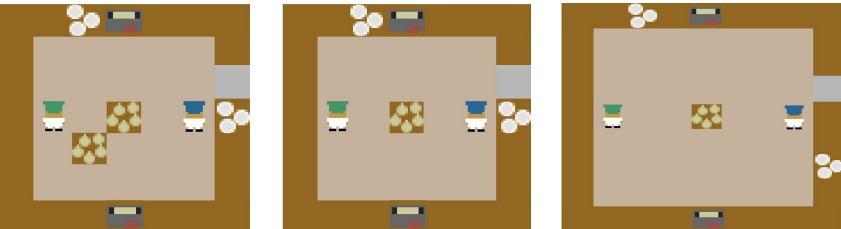

Figure 7: The images of the map of Overcooked tasks. From left to right is easy, medium and hard.

figures. All experiments are carried out on the same computer, equipped with Intel(R) Xeon(R) Gold 5218R CPU @ 2.10GHz, 64GB RAM and an NVIDIA RTX3090. The system is Ubuntu 18.04 and the framework is PyTorch.

## C    DETAILS OF MODEL IMPLEMENTATION AND HYPERPARAMETERS

The network of all compared methods uses the same LSTM network, consisting of a recurrent layer comprised of a GRU with a 64-dimensional hidden state, with one fully-connected layer before and two after. All mixing networks use a fully-connected layer with 32-dimensional hidden state. The network of our critic and policy uses two fully-connected layers with 64-dimensional hidden state and one fully-connected layers with 32-dimensional hidden state after.

## D    PROOF OF THE VALUE DECOMPOSITION OF CRITIC

First of all, according to the decentralized execution setting, there exists a reward decomposition,

$$r_{tot}(s,u) = \sum_{i=1}^{n} r_c^i(o_i, u) = \sum_{i=1}^{n} r_c^i(o_i, u_i^-, a_i). \tag{12}$$

This is because if the task can be solved by decentralized execution, the observation of each agent must contain all the necessary information to identify the goals. Otherwise, agents will require density communication to receive information about others' observations to identify the goals, which is not the setting that we discussed in our works. Then, we define the value decomposition $Q_c^i$ which models each agent's individual utility. From Eq. (12), we have

$$Q_{tot}(s,u) = \mathbb{E}\left[\sum_{t=0}^{\infty} \gamma^t r_{tot}(s,u) \mid \pi\right] = \mathbb{E}\left[\sum_{t=0}^{\infty} \gamma^t \sum_{i=1}^{n} r_{team}^i(o_i, u_i^-, a_i) \mid \pi\right] = \sum_{i=1}^{n} Q_c^i(s,u). \tag{13}$$

In addition, we have

$$\arg\max_{a_i}(Q_{tot}(s,u)) = \arg\max_{a_i}(Q_c^i(s,u)) = \arg\max_{a_i}(Q_c^i(\tau_i, u_i^-, a_i)). \tag{14}$$

The first part is because the value of $a_i$ is represented by item $Q_c^i$ and the reason for the second part is that $Q_c^i$ is only related to agent $i$ as well as the actions of potential cooperative agents and all the necessary information is contained in $(\tau_i, u_i^-, a_i)$, so we can get the unbiased estimated value of $Q_c^i$ given $(\tau_i, u_i^-, a_i)$. Therefore, from Eq. (13) and Eq. (14) we have

$$\arg\max_{u}(Q_{tot}(s,u)) = \{\arg\max_{a_1}(Q_c^1(\tau_1, u_1^-, a_1)), ..., \arg\max_{a_n}(Q_c^n(\tau_n, u_n^-, a_n))\}. \tag{15}$$

An intuitive understanding of Eq. (15) is that each agent takes action based on the perception of other potential cooperative agents' actions, so they can take the corresponding cooperative action and the joint action is the optimal cooperative joint action.

## E    LIMITATIONS OF INDIVIDUAL UTILITY

### E.1    HOMOGENEOUS SCENARIOS

First, for the homogeneous task, we have the payoff matrix in Table 4. Since, we indicate that the

|   | A | B |
|---|---|---|
| A | $+r_1$ | $-r_2$ |
| B | $-r_2$ | 0 |

Table 4: Example of a homogeneous payoff matrix.

|   | A | B |
|---|---|---|
| A | $-r_2$ | $+r_1$ |
| B | $+r_1$ | $-r_2$ |

Table 5: Example of a heterogeneous payoff matrix.

individual utility $Q_i(\tau_i, a_i)$, should be viewed as a variable sampled from distribution $Q_c^i(\tau_i, u_i^-, a_i)$. Following this conclusion, we have the loss of $Q_i(\tau_i, a_i)$ should be

$$\mathcal{L}_i = \sum_{k=1}^{K_i} p_k \cdot (\hat{Q}_c^i(\tau_i, u_i^{k-}, a_i) - Q_i(\tau_i, a_i))^2. \tag{16}$$

where $\hat{Q}_c^i$ means the ground true value function, $u_i^{k-}$ means one of the combination of $u_i^-$ and $p_k$ is the possibility of $u_i^{k-}$ occurred. Therefore, $Q_i(\tau_i, a_i)$ learns to the converged value by optimizing $L_i$, we have the converged $\hat{Q}_i(\tau_i, a_i)$ when $L_i$ is minimized,

$$\hat{Q}_i(\tau_i, a_i) = \sum_{k=1}^{K_i} p_k \cdot \hat{Q}_c^i(\tau_i, u_i^{k-}, a_i). \tag{17}$$

For a simple demonstration, we take the example payoff into Eq. (17). The value of cooperative action $a_i^*$ is

$$\hat{Q}_i(\tau_i, a_i^*) = p_a \cdot \hat{Q}_c^i(\tau_i, u_i^{-*}, a_i^*) + p_b \cdot \hat{Q}_c^i(\tau_i, u_i^-, a_i^*). \tag{18}$$

where $p_a$ means the possibility of other agent taking cooperative actions $u_i^{a-*}$ ($u_i^{a-*} = A$) and $p_b$ means the possibility of other agents taking the other actions $u_i^{b-}$ ($u_i^{b-} = B$). Additionally, we have

$$p_a + p_b = 1 \tag{19}$$

Similarly, we have the value of lazy action $a_i^-$ as

$$\hat{Q}_i(\tau_i, a_i^-) = p_a \cdot \hat{Q}_c^i(\tau_i, u_i^{-*}, a_i^-) + p_b \cdot \hat{Q}_c^i(\tau_i, u_i^-, a_i^-). \tag{20}$$

We know the policy represented by $Q_i(\tau_i, a_i)$ fails when $\hat{Q}_i(\tau_i, a_i^-)$ is larger than $\hat{Q}_i(\tau_i, a_i^*)$, which is

$$\hat{Q}_i(\tau_i, a_i^-) - \hat{Q}_i(\tau_i, a_i^*) = p_a \cdot (\hat{Q}_c^i(\tau_i, u_i^{-*}, a_i^-) - Q_c^i(\tau_i, u_i^{-*}, a_i^*)) \\ + p_b \cdot (\hat{Q}_c^i(\tau_i, u_i^-, a_i^-) - Q_c^i(\tau_i, u_i^-, a_i^*)) > 0 \tag{21}$$

We take the $+r_1$ and $-r_2$ into Eq. (21),

$$\hat{Q}_i(\tau_i, a_i^-) - \hat{Q}_i(\tau_i, a_i^*) = p_a \cdot (-r_2 - r_1) + p_b \cdot (0 - (-r_2)) = (p_b - 1) \cdot (r_2 + r_1) + p_b \cdot r_2 > 0 \tag{22}$$

This means the policy represented by $Q_i(\tau_i, a_i)$ will fail when

$$r_1 \cdot (1 - p_b) < (2p_b - 1) \cdot r_2. \tag{23}$$

which equals to

$$\frac{r_1}{r_2} < \frac{2p_b - 1}{1 - p_b}. \tag{24}$$

### E.2 HETEROGENEOUS SCENARIOS

First, for the heterogeneous task, we have the payoff matrix in Table 5. Similarly, agents with the policy represented by $Q_i(\tau_i, a_i)$ fails when $\hat{Q}_i(\tau_i, a_i^-)$ is larger than $\hat{Q}_i(\tau_i, a_i^*)$ in the heterogeneous scenario. However, there are multiple optimal joint actions (1=A,2=B), (1=B,2=A), which are different from the homogeneous scenarios. We first consider the (1=A,2=B) situation which is

$$\hat{Q}_i(\tau_i, a_i^-) - \hat{Q}_i(\tau_i, a_i^*) = p_b \cdot (\hat{Q}_c^i(\tau_i, u_i^{-*}, a_i^-) - Q_c^i(\tau_i, u_i^{-*}, a_i^*)) \\ + p_a \cdot (\hat{Q}_c^i(\tau_i, u_i^-, a_i^-) - Q_c^i(\tau_i, u_i^-, a_i^*)) > 0 \tag{25}$$

Taking the the $+r_1$ and $-r_2$ into Eq. (25),

$$\hat{Q}_i(\tau_i, a_i^-) - \hat{Q}_i(\tau_i, a_i^*) = p_b \cdot (-r_2 - r_1) + p_a \cdot (r_1 - (-r_2)) \\ = -p_b \cdot (r_2 + r_1) + (1 - p_b) \cdot (r_2 + r_1) > 0 \tag{26}$$

which equals to

$$1 - 2p_b > 0 \tag{27}$$

$$p_b < \tfrac{1}{2}. \tag{28}$$

For situation (1=B,2=A), we have a similar conclusion,

$$p_a < \tfrac{1}{2}. \tag{29}$$

We notice that the overall possibility of failure is

$$P_f = P(p_a < \tfrac{1}{2}) + P(p_b < \tfrac{1}{2}) = P((1 - p_b) < \tfrac{1}{2}) + P(p_b < \tfrac{1}{2}) = P(\tfrac{1}{2} < p_b) + P(p_b < \tfrac{1}{2}) = 1. \tag{30}$$

Therefore, the policy represented by $Q_i(\tau_i, a_i)$ can never promise to solve the heterogeneous task. Furthermore, we can calculate the possibility of reaching cooperation,

$$P_c = P((1 = B, 2 = A)) + P((1 = A, 2 = B)) = p_b \cdot p_a + p_a \cdot p_b = 2 \cdot p_b \cdot (1 - p_b). \tag{31}$$

The maximization of Eq. (31) is 0.5 when $p_b = p_a = 0.5$. The result demonstrated that decreasing $p_b$ when $p_b < 0.5$ causes cooperation more difficult to be reached.

# F  ANALYSIS OF INDIVIDUAL UTILITY OF SEQUENTIAL EXECUTION POLICY

We have the IGM principal as

$$\arg\max_u (Q_{tot}(s, u)) = \{\arg\max_{a_1}(Q_1(\tau_1, a_1)), ..., \arg\max_{a_n}(Q_n(\tau_n, a_n))\}. \tag{32}$$

For sequential execution method, the policy is the form of

$$u = \{\arg\max_{a_1}(Q_s^i(\tau_1, a_1)), ..., \arg\max_{a_n}(Q_s^i(\tau_i, a_{1:n-1}, a_n))\}. \tag{33}$$

We take the payoff matrix in Table 5 as an example, there are multiple optimal joint actions (1=A,2=B), (1=B,2=A), we take the optimal actions into Eq. (33),

$$(1 = A, 2 = B) = \{\arg\max_{a_1}(Q_s^i(\tau_1, a_1)), \arg\max_{a_n}(Q_s^i(\tau_i, A, a_n))\} \\ (1 = B, 2 = A) = \{\arg\max_{a_1}(Q_s^i(\tau_1, a_1)), \arg\max_{a_n}(Q_s^i(\tau_i, B, a_n))\}. \tag{34}$$

We notice that although the latter utility has the correct maximization of the utility, the former one has a conflict maximum result as it lacks the necessary information about other agents' actions.

