# OpenReview forum: "Solving Homogeneous and Heterogeneous Cooperative Tasks with Greedy Sequential Execution"
_ICLR.cc/2024/Conference — ICLR 2024 spotlight_

### Official Review · Reviewer_CxDZ · 2023-10-31

**Soundness:** 2 fair
**Presentation:** 3 good
**Contribution:** 3 good
**Rating:** 6
**Confidence:** 3

**Summary:**

The authors incorporate sequential execution into a value-decomposition-based MARL algorithm, which can thus be applied to both homogeneous and heterogeneous tasks, while maintaining effective credit assignment.
Two approximations are involved: 1) using greedy marginal contribution as the assigned credit for each individual, and 2) using greedy actions (augmented by Monte Carlo samples) as the optimal actions taken by subsequent agents.
The experiments show the efficacy of this approach, demonstrating its ability to adapt to dynamically changing partners within homogeneous, heterogeneous, and mixing scenarios.

**Strengths:**

- Originality. The proposed method addresses common problems in MARL algorithms, such as the relative overgeneralization and credit assignment, at the same time. This enhancement expands the scope of potential applications of value-decomposition-based MARL algorithms.
- Clarity. The paper is basically clear and easy to follow.

**Weaknesses:**

- The algorithm is tested exclusively on specially designed tasks. There are no additional experiments conducted on common benchmarks.
- The ablation study section fails to provide an in-depth discussion of the functions of the key components in the algorithm. I would appreciate a more detailed description of the ablated algorithms and a clearer explanation of the results.

**Questions:**

- To generate a joint action $u$, each agent must calculate its action $a$ sequentially, which prevents parallel processing. Therefore, with each agent requiring $t$ time to produce its action, a single environment step requires $n*t$ units of time for $n$ agents. Compared with baseline algorithms like QMIX, does GSE require more wall-clock time for training and evaluation? If so, could you provide a rough estimate of the additional time required?
- During training, is the execution sequence of all agents fixed, or do you shuffle the sequence?
- What does 'training with a larger scale of agents' mean in Section 6.3 & Figure 6?

---

> ### Author Response · Authors · 2023-11-18
>
> Thank you for your thoughtful questions and feedback on our paper. Below, we provide detailed responses to each of your questions:
>
> 1. The algorithm is tested exclusively on specially designed tasks.
>
>     a. Our work aims to address the challenge of learning both homogeneous and heterogeneous policies simultaneously in value decomposition methods. **Therefore, it is crucial to validate our approach in tasks that require both types of policies.** Unfortunately, existing environments such as SMAC, GRF, and overcooked do not meet this requirement, which necessitated the design of new scenarios.
>
>     b. Furthermore, the tasks encompass a variety of behaviors necessary for completion, and the effectiveness of a task is influenced not just by the environment, but also by the nature of the policy required for its completion. In the environments we have utilized, agents are required to concurrently learn both homogeneous and heterogeneous policies. This dual learning requirement substantially elevates the complexity of these tasks, making them challenging and rich in diversity.
>
> 2. About a more detailed description of the ablated algorithms and a clearer explanation of the results.
>
>     We provide the following clearer explanation of the results in our ablation study and modify the revision accordingly:
>
>     a. **Ablation of Greedy Actions:** Our first ablation test involved evaluating our method without using greedy actions, meaning we relied on marginal contributions instead of greedy marginal contributions. The results indicated a significant performance degradation when greedy actions were not utilized. This finding aligns with our analysis in Section 4.2, where we discussed how using greedy actions helps overcome the relative over-generalization problem.
>
>     b. **Ablation of Marginal Contributions:** The second ablation focused on evaluating our method without marginal contributions. In this scenario, we did not use the critic as proposed in Section 4.1 and directly fitted the policy value function Q_s^i to Q_{tot}. The results demonstrated the challenges of using an implicit credit assignment method to learn the policy when the sequential execution policy does not satisfy the IGM principle, underscoring the importance of our proposed critic.
>
>     c. **Effect of Number of Sampling in MC Method:** We also investigated the impact of varying the number of sampling in the Monte Carlo (MC) method. The findings showed that a small number of samples could lead to suboptimal action selection, failing to identify the greedy value of actions. Conversely, a sufficiently large number of sampling (beyond 5) did not yield additional performance improvements, indicating an optimal range for the number of sampling.
>
>     d. **Performance with Increased Numbers of Agents:** Lastly, we tested our method in the Lift task with double and triple the number of agents. These results affirmed our method's robustness, demonstrating effective performance management even with a larger number of agents.
>
>     We believe these clarifications provide a comprehensive understanding of the key components and functions in our algorithm, as well as the implications of each ablation test conducted.
>
> 3. About the additional time required for training and evaluation.
>
>     It is true that the sequential execution in GSE necessitates more time per step, as each agent calculates its action sequentially. This additional execution time can be estimated as (n-1)*t for n agents, where t is the time taken by each agent to produce its action. However, it's important to note that **this sequential execution primarily impacts the execution phase and not the training phase**. During training, as the transitions are collected, the sequential nature of execution does not impede the parallel processing capabilities. Therefore, the additional time required during training is mitigated. We appreciate the suggestion to focus on minimizing additional execution time. We have included a discussion about this limitation in our work, underscoring its significance. Thank you for this valuable input.
>
> 4. During training, is the execution sequence of all agents fixed, or do you shuffle the sequence?
>
>     We employ a fixed order for the execution sequence of agents, which is determined based on the agent IDs.
>
> 5. What does 'training with a larger scale of agents' mean in Section 6.3 & Figure 6?
>
>     We tested our method in the Lift task with double and triple the number of agents. These results affirmed our method's robustness, demonstrating effective performance management even with a larger number of agents.
>
>
> Thank you once again for your constructive feedback. We believe that your insights have helped us to improve the clarity and quality of our paper.

---

> > ### Comment · Reviewer_CxDZ · 2023-11-21
> >
> > Thank you for your reply. Your clarifications have furthered my understanding of your work. I appreciate your efforts in addressing my concerns during the review process.

---

### Official Review · Reviewer_GHAQ · 2023-11-01

**Soundness:** 3 good
**Presentation:** 4 excellent
**Contribution:** 3 good
**Rating:** 8
**Confidence:** 3

**Summary:**

The paper proposes a way of handling cooperative tasks with multi-agents when the tasks are not all homogeneous.
The approach proposed uses sequential execution policies, proposing the Greedy Sequential Execution (GSE), which learns the optimal policy for both cases (homogeneous and heterogeneous tasks). The GSE is evaluated empirically in multiple domains. The GSE  uses a value decomposition method that works for both homogeneous and heterogeneous tasks, and enables the agents to learn utilities that take into account the interactions with other agents. They also propose a credit assignment method that computes the marginal contributions of each agent. The marginal contribution avoids over-generalization, since it represents the optimal value of an action instead of the average value. A couple of theorems are proved in an appendix (not attached to the paper). The experiments use homogeneous scenarios, heterogeneous scenarios, and mixed scenarios.
Multiple appendices are mentioned but they are not attached to the paper and are not accessible.

**Strengths:**

The paper is very well written and easy to follow. The work presented addresses a known problem (over-generalization) that takes place when combining homogeneous and heterogeneous tasks and expands the applicability of agent-based approaches to situations where some tasks are homogeneous and some are heterogeneous. Having to deal with a mix of the two types of tasks might not be too common, but when mixes of tasks are used, the approach proposed here, even if greedy, becomes quite useful.

**Weaknesses:**

The paper is more appropriate for a journal than for a conference. The numerous appendices are not included in it for space reasons, but are important to understand the method more in depth.
Figure 2 with the architecture is hard to read and not well explained.

**Questions:**

1. The use of Monte Carlo method to estimate the optimal cooperative actions with previous agents to maximize the greedy marginal contribution is mentioned with no other details. How is the number of actions selected?
2. In the Overcooked scenario, I understand why the size of the map affects the complexity, but there is no indication of how large the maps are. I believe the number of agents is fixed to two. Have you tried a larger number of agents?

---

> ### Author Response · Authors · 2023-11-18
>
> Thank you for your thoughtful questions and feedback on our paper. Below, we provide detailed responses to each of your questions:
>
> 1. Figure 2 with the architecture is hard to read and not well explained.
>
>     We have added a more detailed description to accompany Figure 2 in our manuscript. This includes:
>
> - Upper (Blue): A depiction of the critic architecture that we elaborated on in Section 4.1.
> - Lower (Green): An illustration of the framework used for calculating the greedy marginal contribution. This is based on $Q_c^i$ and $V_c^i$, as well as the sequential execution policy $Q_s^i$.
> 2. How is the number of actions of Monte Carlo method selected?
>
>     The number of actions selected for this estimation process is treated as a hyperparameter in our method. In our ablation studies, we have evaluated the effect of different numbers of sampling on the final results. Specifically, we examined how the sample number M affects performance. The results demonstrate that a small value of M can be problematic as it may not select the greedy value of actions. However, a reasonably large value of M is sufficient as increasing M beyond 5 does not further improve performance.
>
> 2. About the size of the map in Overcooked and scenarios with a larger number of agents.
>
>     To provide a clearer understanding of the environment's complexity, we have included more detailed information about the Overcooked scenario and schematic representations of the maps in the appendix. The sizes of the maps used in our experiments are 5x5 for easy and medium and 7x7 for hard.
>
>     Regarding the exploration with a larger number of agents, we have conducted evaluations in the Lift task using double or triple agents. The results of these evaluations are presented in the fourth figure of Figure 6. These results demonstrate that our method effectively handles a larger number of agents, maintaining performance without significant degradation.
>
>
> Thank you once again for your constructive feedback. We believe that your insights have helped us to improve the clarity and quality of our paper.

---

> > ### Comment · Reviewer_GHAQ · 2023-11-22
> >
> > Thank you for answering my questions/comments and for the improvements in the paper.

---

### Official Review · Reviewer_ia9o · 2023-11-01

**Soundness:** 3 good
**Presentation:** 3 good
**Contribution:** 3 good
**Rating:** 8
**Confidence:** 4

**Summary:**

The paper presents a unified framework for learning policies for multiple agents where there are homogeneous and heterogeneous tasks, with the goal of addressing limitations of current methods that work with either homogeneous or heterogeneous tasks. Specifically, the paper proposes greedy sequential execution, with value decomposition including a utility that encodes also the interactions between agents and credit assignment calculated as marginal contribution of Shapley values. Simulation experiments considering different types of tasks are performed and results include comparison with other methods.

**Strengths:**

- the paper addresses an open problem in MARL, where a general framework able to learn optimal policies for both homogeneous and heterogeneous tasks is still not fully present.

- the paper presents a technically sound method, with the augmented utility, sampled considering the joint actions of other agents who might cooperate, and with the greedy marginal contribution

- the paper includes a fairly comprehensive evaluation of the proposed method and compares with a good number of state-of-the-art approaches, with corresponding insights from the results, as well as ablation studies.

- the paper has a structure that overall clearly show the gap, with specific examples, and motivates the proposed approach.

**Weaknesses:**

- a trend that should be discussed more in detail is in Overcooked, where there is a large standard deviation for both Easy and Medium, with Easy having a decreasing return past 900 episodes. It seems that it didn't converge.

A few minor presentation comments:
- Section V already introduces comparison methods that are presented in Section VI. It is better instead to introduce the comparison methods in Section V so that the reader doesn't have to guess what methods are they.
- it is worth including graphically the map for the overcooked environment.
- the references, when the authors name are not used in the sentence should be all in parentheses, e.g., " experiences Sunehag et al. (2017); Rashid et al. (2018)." -> "experiences (Sunehag et al., 2017); (Rashid et al., 2018)."
- please ensure to include the correct venue for papers, instead of just including the arxiv version, e.g., the MAVEN paper was published in NeurIPS 2019.

**Questions:**

Please comment on the trends of overcooked as mentioned in the "Weaknesses" box.

---

> ### Author Response · Authors · 2023-11-18
>
> Thank you for your thoughtful questions and feedback on our paper. Below, we provide detailed responses to each of your questions:
>
> 1. Discuss more in detail in Overcooked.
>
>     The performance in the Overcooked environment is impacted by a variety of factors. **Our settings of difficulty are calibrated based on the ease of discovering rewards.** In simpler scenarios, characterized by more interactive objects and smaller maps, agents have increased opportunities for interaction. **This design, while lowering the barrier for exploration, potentially escalates the difficulty of policy learning as agents progress.** For instance, in scenarios with multiple onions, agents may quickly learn the value of picking up an onion. However, as they develop, they might attempt to pick up different onions, leading to mis-cooperation. This issue is less likely in harder scenarios with fewer onions. These are the reasons that lead to some degree of fluctuation. **However, since the primary challenge of learning is relative overgeneralization and the difficulty in discovering successful instances, our setting is appropriate.** We have added a discussion about this phenomenon in our work. Thank you once again for this suggestion.
>
> 2. It is better instead to introduce the comparison methods in Section V.
>
>     We have reorganized the content to introduce the comparison methods directly in Section V. We believe that this revised order will enhance the readability and coherence of our manuscript, making it easier for readers to follow the logic and rationale behind our comparative analysis.
>
> 3. It is worth including graphically the map for the overcooked environment.
>
>     We have included images of the map in the appendix of our manuscript.
>
> 4. About citation.
>
>     We have revised the citation format throughout our manuscript. We have also updated the manuscript to include the correct publication venues for referenced papers. We appreciate your guidance on these matters and believe these revisions enhance the accuracy and professionalism of our manuscript.
>
>
> Thank you once again for your constructive feedback. We believe that your insights have helped us to improve the clarity and quality of our paper.

---

> > ### Comment · Reviewer_ia9o · 2023-11-20
> >
> > Thanks for the response to this and others' reviews. Overall, the paper appears improved.

---

### Official Review · Reviewer_CQwR · 2023-11-06

**Soundness:** 3 good
**Presentation:** 3 good
**Contribution:** 2 fair
**Rating:** 6
**Confidence:** 5

**Summary:**

The authors propose a general framework for solving both homogeneous and heterogeneous cooperative tasks, which they call Greedy Sequential Execution (GSE). The key idea behind GSE is to sequentially execute the actions of each agent, while taking into account the dependencies between agents measured by greedy marginal contribution.

**Strengths:**

1. The analysis in Sec 3.1 is valuable for readers to gain a deeper understanding of the performance of different policies in homogeneous and heterogeneous scenarios.

**Weaknesses:**

The authors claim that existing solutions have not been successful in **addressing both homogeneous and heterogeneous** scenarios simultaneously, and thus GSE is the first to propose for both scenarios, whose key idea is integrating **sequential execution** and **value decomposition** framework for better **credit assignment** and **cooperation**. However, [1] was published in 2022, which also proposed a general framework based on **sequential execution** for **cooperative games** with **both homogeneous and heterogeneous** agents, and leveraged **advantage value decomposition** for **credit assignment**.

There are too many overlaps in key ideas between these two works, while the main difference seems to be [1] implements these ideas with PPO and sequence model, i.e. transformer, and GSE implements these ideas in a QMIX-like pattern.

However, I have not found any comparison, discussion, or citation to [1] in this paper, which should be compared thoroughly, or the contributions might be significantly weakened.

[1] Wen, Muning, et al. "Multi-agent reinforcement learning is a sequence modeling problem." Advances in Neural Information Processing Systems 35 (2022): 16509-16521.

**Questions:**

I have no further question.

---

> ### Author Response · Authors · 2023-11-18
> **Rebuttal by Authors (1/3)**
>
> Thank you for your suggestion! In the following, we use **MAT** to refer to your suggested work [1]. According to your suggestion, we have now explicitly discussed the relationships between our solution and MAT in the related works of our revision. This discussion also involves HAPPO, which is the base of MAT. We now summarize the differences between our work and these methods for your reference.
>
> The contributions of our method are as follows:
>
> 1. Our proposed utility function adheres to the Individual-Global-Maximum (IGM) principle, with the policy depending solely on local observation.
>
> 2. Our method analyzes and can overcome the problem of relative over-generalization.
>
> 3. Our method achieves explicit credit assignment.
>
> 4. The policy utilizes sequential execution to learn both homogeneous and heterogeneous policies.
>
> In the table below, we compare the characteristics of our method with MAT, HAPPO, as well as our selected baselines in the paper (QMIX, CDS, MAVEN, Shapley). It is evident that compared to baseline methods, HAPPO and MAT have less relevance to our method, as they only use sequential execution to achieve monotonic improvement.
>
> | Method    | IGM principle | Policy relies on local observation | Learn both homogeneous and heterogeneous policies | Overcome relative over-generalization | Explicit credit assignment |
> |-----------|---------------|------------------------------------|---------------------------------------------------|--------------------------------------|---------------------------|
> | GSE       | Yes           | Yes                                | Yes                                               | Yes                                  | Yes                       |
> | QMIX      | Yes           | Yes                                | No                                                | No                                   | No                        |
> | CDS       | Yes           | Yes                                | Yes                                               | No                                   | No                        |
> | MAVEN     | Yes           | Yes                                | No                                                | Yes                                  | No                        |
> | Shapley   | No            | Yes                                | No                                                | Yes                                  | Yes                       |
> | MAT       | No            | No                                 | Yes                                               | No                                   | No                        |
> | HAPPO     | No            | No                                 | Yes                                               | No                                   | No                        |
>
> First, we will explore the distinctions in utility function structures between value decomposition and policy gradient (PG) methods, and the unique challenges each approach presents in multi-agent reinforcement learning.
>
> 1. **Utility Functions Structure:**
>
>     a. **We adhere to IGM principle and only requires local observation, whereas MAT requires joint observations:** In our approach, we introduce a distinctive value decomposition framework implemented by a versatile utility function. This utility function is designed to accurately represent reward function while strictly adhering to the IGM principle. This adherence is crucial as it facilitates more effective credit assignment in multi-agent settings, a key aspect of our research's contribution to value decomposition methods. By adhering to the IGM principle, our utility and policy functions depend exclusively on **local observation** for decision-making. In contrast, MAT adopts policy gradient (PG) approaches, utilizing the transformer's encoder as its value function and the decoder as the policy structure, necessitating **joint observations** to generate actions.

---

> ### Author Response · Authors · 2023-11-18
> **Rebuttal by Authors (2/3)**
>
> 1. **Utility Functions Structure:**
>
>     b. **Unique challenges of different paradigms:** Regarding policy gradient (PG) methods, as we all know, value-based methods and PG methods represent different paradigms in MARL, each with its own features. Value decomposition approaches utilize the IGM principle to model the global maximizer of the joint state-action value function as the combination of individual agent value maximizers, providing a clear and scalable optimization strategy [2][3]. Yet, IGM may not always be applicable, such as in sequential execution scenarios. This limitation is precisely what we have addressed in our work. Conversely, PG methods typically extend single-agent actor-critic algorithms to multi-agent contexts, where the critic is trained on joint observations’ value function. While PG methods like MAPPO demonstrate empirical successes, they struggle with ensuring monotonic improvement, prompting methods like HAPPO, which specifically address this issue to enhance previous PG methods. This distinction in paradigms highlights the unique challenges and methodologies that our research brings to MARL. The table illustrates that our focus is on addressing issues pertinent to value decomposition methods, positioning our work in closer relation and comparison to other methods within this framework.
>
> Next, we will delve into the unique aspects of credit assignment in our method, focusing on how it addresses the issue of relative over-generalization and implements explicit credit assignment techniques.
>
> 2. **Credit assignment:**
>
>     a. **Overcome the problem of relative over-generalization:** Our method introduces a novel approach to credit assignment through the concept of **greedy marginal contribution** to address potential issues of mis-cooperation, often referred to as **the relative over-generalization problem**. This problem typically arises when agents executing earlier in the sequence lack crucial information about the subsequent actions of their counterparts. Our method of greedy marginal contribution employs optimal cooperative actions to calculate marginal contributions in collaboration with preceding agents. This approach directly addresses and compensates for information asymmetry, leading to more effective and accurate credit assignment. In contrast, MAT adopts advantage value decomposition within the structure of a transformer. This design is aimed at executing updates both monotonically and in parallel, thereby enhancing the time efficiency compared to previous methods like HAPPO. However, it does not specifically address the relative over-generalization problem of the value decomposition methods, which is a central concern in our approach. By focusing on the optimal cooperative policy and addressing potential mis-cooperation, our method’s credit assignment differentiates itself significantly from the advantage value decomposition used in MAT.
>
>     b. **Explicit credit assignment:** Our method enables explicit credit assignment by utilizing the marginal contribution of Shapley values. It allows us to compute the credit value of an individual agent's action at a single timestep, which can be applied to reward distribution on recruitment [4] or for pricing [5]. In contrast, MAT cannot compute a credit value for a single observation and action pair, as the utility is based on joint observations. Therefore, our solution offers broader applicability in more realistic settings.
>
> Next, we will discuss the proposal and contributions of using sequential execution in different methods.
>
> 3. **Propose of using sequential execution:**
>
>     a. **Different objectives:** Our work delves into why existing value decomposition methods fail to simultaneously represent both homogeneous and heterogeneous policies. Through this analysis, we introduce the concept of **greedy sequential execution** as a novel solution. The core contribution of our research lies not merely in proposing a new sequential execution method, but in innovatively applying this concept within the context of value-based methods by proposing an IGM utility function as well as the greedy sequential execution to solve the relative over-generalization problem. These problems have not been analyzed and solved before and are important for the value decomposition method to be implemented in real-world scenarios. In contrast, MAT does not directly address the challenge that **merely using sequential execution policies does not conform to the IGM principle and cannot solve the relative over-generalization problem**. This is because MAT leverages transformer structures to achieve on-policy and monotonic performance enhancement within the framework of PG methods, without specifically addressing the challenges we focus on. Therefore, MAT is not suitable to be compared as a baseline in our setting.

---

> ### Author Response · Authors · 2023-11-18
> **Rebuttal by Authors (3/3)**
>
> In summary, while both our method and MAT aim to improve the performance of cooperative multi-agent systems, they do so from different perspectives and using different techniques. We believe that our approach offers a novel and effective solution to the challenges of learning both homogeneous and heterogeneous policies simultaneously when using value decomposition methods in multi-agent systems. Its efficiency is further validated through comparisons with other value decomposition methods.
>
> Thank you once again for your constructive feedback. We believe that your insights have helped us to improve the clarity and quality of our paper.
>
> [1] Wen, Muning, et al. "Multi-agent reinforcement learning is a sequence modeling problem." Advances in Neural Information Processing Systems 35 (2022): 16509-16521.
>
> [2] Dou Z, Kuba J G, Yang Y. Understanding value decomposition algorithms in deep cooperative multi-agent reinforcement learning[J]. arXiv preprint arXiv:2202.04868, 2022.
>
> [3] Wang J, Ren Z, Liu T, et al. QPLEX: Duplex Dueling Multi-Agent Q-Learning[C]. International Conference on Learning Representations. 2020.
>
> [4] Shen S, Ji M, Wu Z, et al. An optimization approach for worker selection in crowdsourcing systems[J]. Computers & Industrial Engineering, 2022, 173: 108730.
>
> [5] Landinez-Lamadrid D C, Ramirez-Ríos D G, Neira Rodado D, et al. Shapley Value: its algorithms and application to supply chains[J]. INGE CUC, 2017, 13(1): 53-60.

---

> ### Comment · Reviewer_CQwR · 2023-11-20
> **Thanks so for authors' clarification and here are a few further suggestions.**
>
> Thank you for the clarification, which makes the contributions of this work more clear and specific, especially the 2.a & 2.b.
>
> Just a personal thought, the **Explicit credit assignment** with greedy marginal contribution is the most attractive contribution of this work in the field of value-decomposition methods for cooperative MARL. But the original writing spent too much in emphasizing **handling both homogeneous and heterogeneous** (actually, it is not a new topic currently), which instead led to obscuring the contribution of the explicit credit assignment as well as locating this work into an unnecessarily over-large scope. In other words, if this work mainly claims that "we propose a new framework that addresses both homogeneous and heterogeneous scenarios well", you have to compare it with other existing works that address both homogeneous and heterogeneous scenarios like the MAT to prove this work is non-trivial (since from the title of the article, the MAT can also be described similarly as "SEQUENTIAL EXECUTION: SOLVING HOMOGENEOUS AND HETEROGENEOUS COOPERATIVE TASKS WITH A UNIFIED FRAMEWORK"). Otherwise, if the main claim of this work is a novel value-decomposition method that is capable of both homogeneous and heterogeneous agents, the comparison is unnecessary but the scope of contribution in the original writing should be adjusted and explained a bit according to the rebuttal provided by authors (this rebuttal is more specific about the scope of this work then that in the original writing).
>
> It is overall a writing issue instead of a technical issue, if authors can make sure to address my concerns in the later version, I am willing to update my score to positive.
>
> Minor issue:
> Similar reason to the above discussion, the beginning sentences in the abstract "Effectively handling both homogeneous and heterogeneous tasks is crucial for the practical application of cooperative agents. However, existing solutions have not been successful in addressing both types of tasks simultaneously." are misleading without a correct specification about the scope of this paper. (Is MAT an existing solution in addressing both homogeneous and heterogeneous tasks? Is it successful? If not, why?)

---

> > ### Author Response · Authors · 2023-11-21
> >
> > Thank you for your insightful feedback on our manuscript. Based on your suggestions, we have made several adjustments to our paper, highlighted in red in the revised paper for clarity:
> >
> > 1. **Title**: We have changed the title to "Solving Homogeneous and Heterogeneous Cooperative Tasks with Greedy Sequential Execution." This revision emphasizes our method, Greedy Sequential Execution (GSE), more prominently.
> > 2. **Abstract**: The abstract has been restructured to highlight our contributions to value decomposition methods, specifically addressing credit assignment and relative over-generalization challenges, ensuring a clear and precise representation to avoid any misunderstanding. We have specifically highlighted the shortcomings of the sequential execution method and how GSE overcomes these issues.
> > 3. **Introduction Section**: The introduction now more explicitly discusses the challenges faced by current value decomposition methods and the limitations of directly using sequential execution. We emphasize how GSE contributes to addressing these issues simultaneously.
> > 4. **Motivating Examples Section**: This section has been revised to underline the limitations of value decomposition utility. We have also reordered the analysis of sequential execution to be consistent with the earlier discussion in the paper.
> > 5. **Method Section**: We have refocused the introduction of the method section on solving the problems of credit assignment and relative over-generalization.
> >
> > We believe these revisions more accurately reflect the core contributions of our work and address the concerns you've raised. We hope these changes make our paper clearer and more focused on its novel contributions to the field of MARL.

---

> > > ### Author Response · Authors · 2023-11-22
> > >
> > > As the deadline is nearing, we hope this latest revision meets your requirements. Should you need any further adjustments, we remain ready to continue refining our work.

---

> ### Comment · Reviewer_CQwR · 2023-11-22
> **Thanks for authors' effort**
>
> Thanks for providing the revision, this revision is clearer about the scope of this paper and has addressed most of my concerns. I am delighted to increase my score.

---

> > ### Author Response · Authors · 2023-11-22
> >
> > Thank you for your reply. We greatly appreciate your effort during our discussions.

---

### Meta-Review · Program_Chairs · 2023-12-09

**Metareview:**

Synopsis: This paper develops an algorithm for learning policies effectively in both homogeneous, as well as heterogeneous cooperative task settings. It proposes Greedy Sequential Execution as a solution to this problem, and empirically demonstrates improvements over baselines in both homogeneous and heterogeneous settings.

Strengths:
+ Tackles an important open problem in multiagent reinforcement learning
+ Clear technical presentation, with plenty of insights and examples to lead the reader through the approach
+ Good empirical results

Weaknesses:
- The original paper missed a key reference and comparison, but this has since been addressed by the authors' revisions - much appreciated.

**Justification For Why Not Higher Score:**

The paper addresses a very specific task setup, that potentially is of narrow scope for an ICLR audience.

**Justification For Why Not Lower Score:**

There is a fair bit of technical depth, and it tackles an important problem in a non-trivial manner, with good results.

---

### Decision · Program_Chairs · 2024-01-16

Accept (spotlight)